# Uncovering the Secrets of Prostate Cancer’s Radiotherapy Resistance: Advances in Mechanism Research

**DOI:** 10.3390/biomedicines11061628

**Published:** 2023-06-03

**Authors:** Feng Lyu, Shi-Yu Shang, Xian-Shu Gao, Ming-Wei Ma, Mu Xie, Xue-Ying Ren, Ming-Zhu Liu, Jia-Yan Chen, Shan-Shi Li, Lei Huang

**Affiliations:** 1Department of Radiation Oncology, Peking University First Hospital, Beijing 100034, China; lyufeng2018@bjmu.edu.cn (F.L.); ssy970726@163.com (S.-Y.S.); 15810160120@163.com (M.-W.M.); xiemu0312@126.com (M.X.); xy_ren1031@126.com (X.-Y.R.); liumingzhu125@126.com (M.-Z.L.); sunshinecan2011@sina.com (J.-Y.C.); shanshili98@163.com (S.-S.L.); mp_huanglei@163.com (L.H.); 2First Clinical Medical School, Hebei North University, Zhangjiakou 075000, China

**Keywords:** prostate cancer, radiation resistance

## Abstract

Prostate cancer (PCa) is a critical global public health issue with its incidence on the rise. Radiation therapy holds a primary role in PCa treatment; however, radiation resistance has become increasingly challenging as we uncover more about PCa’s pathogenesis. Our review aims to investigate the multifaceted mechanisms underlying radiation therapy resistance in PCa. Specifically, we will examine how various factors, such as cell cycle regulation, DNA damage repair, hypoxic conditions, oxidative stress, testosterone levels, epithelial–mesenchymal transition, and tumor stem cells, contribute to radiation therapy resistance. By exploring these mechanisms, we hope to offer new insights and directions towards overcoming the challenges of radiation therapy resistance in PCa. This can also provide a theoretical basis for the clinical application of novel ultra-high-dose-rate (FLASH) radiotherapy in the era of PCa.

## 1. Introduction

Prostate cancer (PCa) is the second most common malignant tumor in men worldwide and the fifth leading cause of death. According to the latest epidemiological survey results, PCa was the third most commonly diagnosed malignancy in 2020, with 1,414,259 new cases (7.3% of total) [1], following only lung and colorectal cancers. Prostate-specific antigen (PSA) is a prevalent biomarker employed for the diagnosis and active surveillance of PCa. According to the NCCN guidelines, PSA plays a crucial role in determining PCa risk stratification. Specifically, a PSA level of less than 10 ng/mL indicates low-risk PCa, 10 to 20 ng/mL suggests intermediate-risk PCa, and levels exceeding 20 ng/mL are indicative of high-risk PCa. Consequently, PSA screening is an important component for the diagnosis of PCa [2]. PCa exhibits significant heterogeneity between different patients, resulting in varying treatment responses and outcomes. Therefore, a detailed analysis of the underlying molecular mechanisms is necessary to achieve personalized diagnosis and treatment.

In 1917, Professor Benjamin Barringer of Memorial Sloan-Kettering Cancer Center in the United States pioneered the use of radium radiation to treat PCa, and since then radiation therapy has become a common treatment option for PCa. In 1962, Professor Bagshaw of Stanford University in the United States pioneered high-dose radiotherapy technology (Appendix A) and subsequently applied it as a radical treatment for PCa, conducting numerous case studies that established the status of radiotherapy as one of the “Troika” of PCa treatment [3,4]. Radiotherapy has been clinically applied to all stages of PCa, including localized, metastatic, and castration-resistant PCa (CRPC). While radiotherapy is considered the standard treatment for localized PCa, its effectiveness is comparable to that of radical prostatectomy (RP) surgery. As a result, the use of radiation therapy in PCa has been recommended as a treatment option in widely recognized guidelines worldwide, such as those from the National Comprehensive Cancer Network (NCCN) and the European Association of Urology (EAU). However, some clinical studies have demonstrated that there is resistance to radiotherapy among certain PCa patients [5,6]. For a long time, the issue of radiotherapy resistance in PCa was overlooked. As tumor resistance improves, bladder and rectal toxicities increase during the course of radiotherapy, making treatment more challenging. This necessitates a gradual increase in the ionizing radiation (IR) dose, which can be a dilemma for patients. Therefore, it is critical to enhance PCa’s response to radiotherapy, with the ideal strategy being the discovery of better-tolerated drugs to enhance radiotherapy’s efficacy and permit a safe and effective increase in dosage.

With the advent of personalized and precision medicine, a comprehensive and in-depth understanding of the mechanism of radiotherapy resistance in PCa is essential. Therefore, the development of radiosensitization drugs targeting possible resistance mechanisms may provide novel ideas about and solutions to the clinical radiosensitization of PCa. Current research suggests that radiotherapy resistance in PCa is a complex process involving various factors, multiple genes, multiple pathways, and multiple mechanisms. In the following sections, we will describe and summarize the possible mechanisms of radiotherapy resistance in PCa.

## 2. Impact of Cellular Hormone Levels on PCa

Current androgen deprivation therapy (ADT) was established based on the groundbreaking work of Huggins and Hodges in 1941 [7,8], where they discovered that reducing androgen levels could slow the progression of metastatic PCa. Recent studies have demonstrated that androgen biosynthesis is tightly regulated by the “hypothalamic-pituitary-gonad (HPG)” axis [9], and that androgen receptor (AR) axis signaling plays a crucial role in the onset and progression of PCa [10,11]. Androgens, such as testosterone and dihydrotestosterone, bind to the ligand-binding domain (LBD) of AR in cells, causing AR to detach from HSP90 and move into the nucleus, where it interacts with androgen response elements (AREs) to activate the expression of genes such as KLK3, NKX3.1, FKBP5, and TMPRSS2-ERG, among others. It is believed that malignant transformation in PCa is driven by this fundamental physiological process [12,13].

As CRPC is the final clinical stage of ADT, CRPC cells are more prone to resistance towards multiple treatment regimens (such as ADT therapy, radiotherapy, and chemotherapy) compared to hormone-sensitive PCa cells [14,15]. There appears to be a correlation between hormone level and DNA damage repair, as studies have shown that after ADT therapy, the expression of Ku70 (the key junction protein of non-homologous end joining (NHEJ)) in PCa tissue is downregulated following DNA damage [16]. Furthermore, recent clinical research has also demonstrated that the local failure rate of CRPC after local treatment is significantly higher than that of hormone-sensitive PCa [6], indicating that the radiosensitivity of PCa is reduced when the PCa is transformed from an androgen-dependent state to an androgen-independent state. In terms of drug mechanism, theoretically, ADT could have a synergistic effect with radiotherapy for PCa because it can induce the apoptosis and autophagy of PCa cells [17]. In addition, androgen-sensitive PCa LNCaP cells expressing AR undergo a higher rate of radiation-induced apoptosis in the absence of androgens [18], which is consistent with findings from other studies. It is worth noting that PCa cells expressing AR experience the same response. Increasingly deepening research suggests that AR, as a receptor of steroid hormones, has a causal and mutually regulated association with radiotherapy resistance in PCa due to its similarity of location and spatial accessibility to DNA. In PCa cells, DNA damage caused by radiation can activate AR, and after AR activation, DNA PKcs’ activity and expression are induced (a significant molecule of NHEJ). Furthermore, a positive feedback regulatory loop involving AR and DNA PKcs can be formed [19,20,21] as shown in Figure 1.

In addition, patients with metastatic CRPC (mCRPC) are often accompanied by mutations and amplification of the AR, as well as the production of splice variants of AR. Mutations and amplifications result in the lack of the LBD during mRNA splicing, leading to the retention of the N-terminal AF1 activation cluster (N-TAD) and the DNA-binding domain (DNBD), forming a splice variant of AR [22,23]. This allows for the aberrant activation of the AR signaling pathway inside the cell, even under conditions of castrated androgen in the extracellular environment. This mechanism is involved in the occurrence and progression of drug resistance and castration resistance and is considered one of the challenges restricting the improvement of ADT effectiveness in clinical practice [13]. The most studied AR splice variant (ARv) in endocrine therapy is androgen receptor variant 7 (AR-V7). AR-V7 plays a similar role to AR in the radiotherapy resistance of PCa. After 5 Gy of radiotherapy, AR-V7 expression was upregulated in PCa C4-2 and 22Rv1cells, and their nuclear translocation was elevated. AR-V7 can promote DNA damage response (DDR) and initiate the repair process of homologous recombination repair (HRR) and NHEJ. The upregulated expression of AR-V7 after IR can reduce the “synthetic lethality” response of poly (ADP-ribose) polymerase 1 (PARP-1) inhibitor [24]. Furthermore, by binding to the key molecule of NHEJ, DNA PKcs, AR-V7 can form a DNA repair complex and enhance the repair ability of cells after radiation, thereby fostering the radiotherapy resistance of PCa cells, which can be inhibited by the AR antagonist enzalutamide [25].

## 3. Maintaining DNA Repair Deficiency and Disorders of the DNA Damage Repair System

One of the mechanisms underlying X-ray therapy for cancer is damaging the DNA in tumor cells. Currently, the mechanisms leading to the death of tumor cells are as follows: direct damage to the DNA target of tumor cells, related factors, and molecules causing various programmed deaths (e.g., apoptosis, autophagy, programmed necrosis, and senescence) that occur in tumor cells due to cellular communication leading to their induction, inducing cells to undergo mitotic death, etc. DNA serves as an essential target for radiation-induced death, and various types of radiation-induced DNA damage can be recognized through a complex network of pathways. Corresponding repair processes for different types of DNA damage can also be initiated to maintain genomic stability [26]. When tumor cells are irradiated, DDR is crucial for activating repair pathways and cells, and DDR receptor proteins that respond to multiple DNA damages are critical for initiating repair. Different molecules and mechanisms involved in damage repair play distinct roles in the radiotherapy resistance of PCa (Table 1 and Appendix A).

Hence, due to the complex nature of the DNA damage repair process involving multiple molecules and complexes with intricate mechanisms, it is necessary to conduct further studies on the role of these complex components in promoting radiotherapy resistance in PCa. Furthermore, there is a need to develop corresponding drugs targeting specific molecular targets to enhance radiosensitivity in clinically high-grade malignant PCa.

## 4. Cell Cycle Disorder

According to classical radiobiology theory, the radiosensitivity of cells is determined by the “4R” factors: repair (sub-lethal injury and potentially lethal injury), repopulation, redistribution (cell cycle), and re-oxygenation. The cell cycle precisely regulates cellular activities that determine cell proliferation and fate after DNA damage, which is finely controlled by a series of cell cycle proteins and corresponding kinases. Abnormal regulation of the cell cycle is a hallmark of tumor cells [52]. The G1/S and G2/M checkpoints play vital roles in regulating the entire cell cycle, including determining entry into the DNA synthesis phase and proper cell division. As the primary target of IR, DNA damage repair is closely linked to cell cycle regulation in PCa radiotherapy resistance. Cyclins, kinases, and inhibitors are key regulatory molecules of the cell cycle and represent critical targets for current malignant tumor treatment strategies. Notably, CDK4/6 inhibitors have shown potential as a treatment approach for enhancing radiosensitivity against malignant tumors. The successful application of this approach not only offers a possible treatment method from the perspective of the cell cycle but also sheds new light on the precise control of radiosensitivity in PCa. Current research on using cell cycle regulatory molecules as targets for improving radiosensitization in PCa is summarized below (Table 2 and Appendix A).

The cell cycle is a precisely regulated machinery with intricate details and processes. In PCa, various molecules can enhance radiosensitization through different processes, molecules, and mechanisms. However, their corresponding targets and underlying mechanisms require further exploration to fully understand the intrinsic correlation between cell cycle regulation disorder and radiotherapy resistance in PCa. A deeper understanding of these mechanisms will facilitate the development of more effective treatment strategies.

## 5. Disruption of Cellular Redox Homeostasis

Maintaining cellular redox homeostasis is essential for normal cell proliferation, signal transduction, and physiological activities. Tumor cells heavily rely on maintaining stable levels of reactive oxygen species (ROS) as well. During tumor initiation and progression, the rapid and excessive proliferation of cancer cells results in the production of massive amounts of ROS which damage tumor cell DNA and lead to genomic instability [61]. During malignant tumor invasion and metastasis, ROS not only promotes tumor cell proliferation, but also interacts with stromal and immune cells to activate the EMT-related TGF-β pathway and transcription factors, leading to tumor cell scattering and distant metastasis.

Radiotherapy can have both direct and indirect ionization effects when used to treat cancer. One indirect effect of IR is the generation of ROS in tumor cells through interactions between oxygen and water molecules. These ROS include superoxide anion, hydrogen peroxide, hydroxyl radicals, singlet oxygen, and others [62]. Additionally, IR can induce cells to produce reactive nitrogen species (RNS), including nitric oxide (NO), and other oxides or nitrogen-containing free radicals. These chemical species are highly reactive, unstable, and paramagnetic compared to other molecules found in nature. They can disrupt the structure and function of biomolecules such as DNA, proteins, and lipids and also disrupt the redox homeostasis of tumor cells. This disruption can trigger damage repair, apoptosis, autophagy, and ferroptosis in tumor cells, making them useful in treating tumors [63].

Numerous studies have demonstrated that approximately 70% of IR’s therapeutic effects are due to its indirect ionization effects [64,65]. During tumor evolution and subsequent treatment, the organism may develop adaptive mechanisms to resist oxidative stress (Figure 2). However, an overactive adaptive mechanism can lead to cellular tolerance to oxidative stress damage and subsequent treatment resistance. Emerging evidence suggests that multiple adaptive mechanisms of antioxidant stress in tumor cells are at play, including metabolic reprogramming via sulfur-based metabolism to produce antioxidant substances, weakening the metabolism of glutamate and folic acid, enhancing the metabolism of the pentose phosphate pathway to increase production of NADPH, enhancing the transcription and expression of antioxidant-stress-related transcription factors and genes, and stimulating the metabolic signaling pathway involving AMPK. Therefore, it is crucial to study the molecular mechanisms that control redox homeostasis in tumor cells to develop effective therapies.

The organism’s antioxidant stress defense system is composed of antioxidant enzymes, antioxidant substances, and antioxidant-stress-related transcription factors, all of which play a crucial role in maintaining cellular redox homeostasis. One such transcription factor is nuclear factor erythroid 2-related factor 2 (Nrf2) [66], a member of the leucine zipper transcription factor family that typically resides in the cytoplasm and is continuously ubiquitinated and degraded by Kelch-like ECH-associated protein 1 (Keap1) to maintain low expression levels. When cells undergo oxidative stress, endogenous oxidative stress inducers (ROS and metabolites produced during the cell’s oxidative phosphorylation process) and exogenous oxidative stress inducers (IR and chemotherapy drugs) are involved in the cell. The antioxidant response element (ARE) on the nucleus is activated by oxidative stress, prompting the transcription factors Nrf2 and Keap1 to depolymerize and translocate to the nucleus, where they form a heterodimer with Maf protein, which is bound to the ARE. This complex then functions to clear oxidative substances from the cell and protect its structure and function by generating antioxidant substances such as glutathione (GSH) and antioxidant enzymes such as catalase (GSH, HO-1, NQO-1, and GPX) [67,68].

The malignant biological behavior of tumors is largely associated with regulatory abnormalities in the pathway responsible for maintaining redox homeostasis, which is a key regulator. In tumor cells, the constant mutation of Keap1 leads to the unregulated expression and localization of Nrf2 [69]. Abnormally high levels of antioxidant stress in tumor cells contribute to proliferation, invasion, metastasis, and therapeutic resistance [70]. The “Nrf2/ARE” signaling pathway has been associated with radiotherapy and chemotherapy resistance in lung cancer (Figure 3) [71,72]. Clinical samples of PCa have revealed three highly methylated sites (H3K9me3, MBD2, and MeCP2) in the Nrf2 promoter region, which inhibits Nrf2 transcription and downregulates PCa cells [73]. Knocking out Nrf2 in the transgenic mouse model of PCa leads to depleted glutathione S-transferase (GST) and increased ROS levels, promoting the PCa development process [74]. Knocking down Nrf2 in PCa DU145 cells can decrease the expression of oxidative-stress-related genes, such as NAD(P)H:Quinone oxidoreductase 1 (NQQ1), superoxide dismutase 2 (SOD2), and heme oxygenase-1 (HO-1), making the cells more sensitive to cis-platinum and inducing DNA damage response [75]. The activation of Nrf2 can induce tolerance to radiation treatment in glioma cells. The overexpression of HECT and Copper Zinc Superoxide Dismutase Domain containing protein 1 (HACE1) in glioma tissues competes with Keap1 to prevent Nrf2 from being degraded via ubiquitination at the post-translational modification level and promotes the upregulation of Nrf2 transcriptional expression via the internal ribosome entry site (IRES) through La/SSB [76]. Targeting Nrf2 can induce ferroptosis in PCa cells [77], providing novel insight into the radiotherapy resistance of PCa. Therefore, Nrf2 is a critical target for improving treatment resistance in PCa [78,79]. However, its role in radiotherapy resistance in PCa remains unclear and requires further study.

Tumor cells reprogram their metabolism, which can differ from normal cells [80,81]. Internal chemical metabolism is central to oxidative stress. Thus, the metabolic reprogramming process of tumor cells not only sheds new light on interpreting radiotherapy resistance in PCa but also offers a novel strategy for radiosensitizing PCa. Tumor cells have an abnormal increase in glutamine catabolism, which rapidly supplies fuel for cell division. Glutaminase-driven catabolism can increase intracellular antioxidant substances such as GSH to protect against IR damage. The increase in glutamine catabolism in PCa cells is not only related to defense against oxidative stress but also involved in maintaining PCa cells’ survival and inducing ATG5-mediated cytoprotective autophagy [82]. Additionally, the detection of glutamine in the peripheral blood, glutaminase 1 (GLS1), and myelocytomatosis viral oncogene homolog (MYC), which regulate glutamine catabolism, can improve screening for the population that will benefit from radiotherapy and predict PSA doubling time in clinical settings. In normal prostate epithelial cells, the glucocorticoid betamethasone activates the “RelB-BLNK” axis, promotes the transcription of manganese superoxide dismutase (MnSOD) after radiation, and protects normal cells from radiation damage. Betamethasone inhibits the “Rel-BLNK” axis, which can further increase ROS in cells, leading to the death of PCa cells [83]. This is necessary due to the high level of ROS in the metabolic reprogramming of PCa cells. The GSH/GSSG ratio is downregulated in PC-3 cells in PCa because parthenolide deploys NADPH oxidase, which uses up thioredoxin. Radiosensitization is achieved by inhibiting PCa cell metabolism [84], which leads to a downregulation of forkhead box O3a (FOXO3a) expression and its downstream molecular antioxidant SOD.

AR and PCa cell redox homeostasis are interconnected, regulating and influencing each other. Currently, ADT can induce oxidative stress damage to PCa cells in addition to targeting AR, leading to therapeutic effects [85,86]. The treatment of PCa cells with ADT results in the induction of endocrine resistance and radiotherapy resistance, as evidenced by an increase in the expression of Nrf2 and antioxidant stress molecules (peroxiredoxin-1, thioredoxin 1, and metallothionein-1) [87]. Under conditions of oxidative stress, thioredoxin domain-containing protein 9 (TXNDC9), the primary regulator of reactive oxygen species, and PRDX1 can become dissociated. By interacting with AR, peroxiredoxin-1 (PRDX1) blocks its ubiquitination degradation, increases AR expression, and maintains AR signaling pathway activation [88]. The disruption of homeostasis in the defense system against oxidative stress varies across tumor stages and processes. Due to this, the role of oxidative stress in the malignant transformation and treatment resistance of malignant tumors needs more research to determine the threshold and detailed role.

## 6. Enhance of Epithelial–Mesenchymal Transitions (EMT)

EMTs refer to the morphological transformation of epithelial cells into mesenchymal cells or fibroblasts, including the disappearance of cell polarity and rearrangement of the cytoskeleton. The enhanced migratory capacity of cells during the EMT process allows for its categorization into three distinct subtypes, which are linked to various biological processes including tumor invasion, migration, metastasis, tumor microenvironment, and immune microenvironment changes [89,90,91]. During the EMT process, epithelioid cell markers, such as E-cadherin, β-catenin, Cladin-1, and zona occludens-1 (ZO-1), are downregulated, while the markers of mesenchymal cells, such as vimentin, α-smooth muscle actin (α-SMA), Snail1/2, Twsit1/2, ZEB1/2, and other molecules, are upregulated. These molecular changes cause complex regulatory network changes within cells, inducing the process of EMT. Inducing factors of EMT include activated intracellular EMT-related pathways (TGF-β/Smad, ERK, NF-κB, Wnt/β-Catenin, and Notch) in interaction with growth factors in the extracellular matrix (ECM) and receptors on the cell membrane surface, interaction between tumor cells and interstitial cells in the tumor environment, and an expression of EMT molecules induced by hypoxia in the tumor.

According to typical radiobiology theory, epithelial tumor cells exhibit moderate sensitivity to IR, while stromal cells are relatively radiation-resistant. Consequently, the process of EMT in tumor cells also contributes to radiation tolerance. In PCa cells, activation of the “acetylated KLF5/CXCR4” axis can induce interleukin-11 (IL-11) secretion, trigger the SHH/IL-6 paracrine pathway, promote docetaxel resistance, and sustain EMT [92]. IR has been shown to increase EMT markers (uPA, vimentin, and N-cadherin) in PCa DU145 cells [93]. EMT markers (N-cadherin and vimentin) were upregulated, while E-cadherin and cofilin expression was downregulated in samples taken from patients with PCa before and after radiation therapy. Together with PARP-1, these markers serve as predictors of PCa’s susceptibility to radiotherapy. The dynamic changes in the EMT-MET process of PCa patients receiving radiotherapy were evaluated and found to be inhibited by silymarin [93]. Lysyl oxidase-like 2 (LOXL2) is a molecule associated with radiotherapy resistance. In PCa cells, knockdown of LOXL2 inhibits the EMT process of CRPC DU145 and PC-3 cells, thereby restoring radiosensitivity [94]. ZEB1, a transcription inhibitor that promotes the EMT process and stemness characteristics, significantly contributes to regulating cell response to radiation. Studies have shown that ZEB1 is upregulated in radiation-resistant cells. ATM phosphorylates and stabilizes ZEB1 expression in response to DNA damage. ZEB1 can directly interact with USP7, enhancing its ability to deubiquitinate and stabilize CHK1 [95], leading to HR-dependent DNA repair and promoting radiation resistance. miR-875-5p targets the epidermal growth factor receptor (EGFR)/ZEB1 signaling pathway, promoting PCa cells’ transition from the EMT process to the MET process, thereby restoring their radiosensitivity [96]. Bissalicylic acid inhibits the EMT process of PCa PC-3 cells. Further research has found that bissalicylic acid synergizes with radiotherapy to sensitize PCa by activating the AMPK pathway and inhibiting the acetyl-CoA carboxylase (ACC) and thioredoxin domain-containing protein 1 (TROC1) pathways, suggesting its potential as a promising cancer therapy in the future [97]. Relevant research is needed to confirm the therapeutic effect of salicylic acid combined with radiotherapy for PCa. Currently, radiotherapy has become one of the treatment methods for metastatic PCa, which broadens the implications of radiotherapy for this disease.

EMT is also associated with molecularly divergent subtypes and aberrant histologies. Variant histologies (VHs) have been recognized as drivers of biological heterogeneity and increased aggressiveness in current clinical practice. In non-muscle-invasive bladder cancer (NIMBC), variant histologies (nested, glandular, micropapillary, squamous, inverted, basaloid, microcystic, villous-like, and lymphoepithelioma-like carcinoma) have been identified as risk factors for patient disease-free survival (DFS) [98]. Plasmacytoid, small-cell, and sarcomatoid VHs are linked to worse disease-specific survival (DSS) in muscle-invasive bladder cancer (MIBC), while lymphoepithelioma-like VH is associated with an improved DSS [99]. An accurate pathological diagnosis of VHs can enable tailored counseling to identify patients who require more intensive management [100]. In addition, ductal adenocarcinoma (DAC) is the most common variant histological subtype of PCa and is characterized by an aggressive clinical course. Recent studies suggest that DAC requires external beam radiation therapy and particle-enhanced therapy, indicating DAC’s resistance to radiation therapy [101,102]. Intraductal carcinoma of the prostate (IDC-P) is positively correlated with higher GS and is associated with early relapse and metastasis after radiation therapy, suggesting IDC-P’s insensitivity to radiation therapy [103,104]. Sarcomatoid carcinoma is also rare and carries a poor prognosis, with limited clinical interventions and approximately 38% of patients experiencing distant metastasis [105]. It most frequently emerges after radiation for a high-grade acinar carcinoma [106]. Some sarcomatoid carcinomas lack classical epithelial features [107], which could be one of the reasons why these patients are resistant to chemotherapy and radiation therapy, leading to a poor prognosis. Moreover, pleomorphic giant-cell adenocarcinoma is a rare and aggressive subtype that often develops following prior treatment with androgen deprivation or radiation [108]. Therefore, these variant histologies are not only strongly associated with risk stratification and survival outcome in patients but also pose a significant challenge in understanding the relationship and mechanisms between ionizing radiation and specific pathological types in PCa.

Since the biological behavior of metastatic PCa is different from that of primary PCa, the cells of these metastatic foci are mostly formed through the EMT process from the primary foci. Research into radiotherapy for both the primary and metastatic foci of metastatic PCa is still in the exploratory stage, with a lack of focus on radiation resistance being a major issue. We anticipate that as research into radiation for PCa continues to advance, treatment for these metastases will become more individualized and accurate.

## 7. The Existence of Prostate Cancer Stem Cells (PCSCs) in Foci

Initially identified in leukemia [109], cancer stem cells (CSCs) are a functionally distinct population of tumor-resident cells characterized by their capacity for self-renewal, differentiation into many cell types, and potential for metastasis [110]. Despite spending much of their time in the G0 phase of the cell cycle, CSCs are resistant to radiation and conventional chemotherapy. Modern clinical research has shown that not all PCa patients exhibit the same biological behaviors despite sharing the same disease [111]. Moreover, the subclonal origin of CSCs and PCSCs may be directly associated with the presence of distinct subgroups exhibiting malignant biological characteristics.

In 2005, PCSCs were identified for the first time in PCa samples collected following RP surgery; these cells exhibited surface markers consistent with traditional CSCs (CD44+, CD133+, EPCaM, ALDH1, Snail, etc.) and were found to have proliferated and differentiated in a manner characteristic of stem cells. PCSCs were subsequently identified in metastatic PCa and chemo- and radiotherapy-resistant tissue samples [112]. CSCs are primarily found in tumor microenvironments. Another self-protective mechanism for CSCs to evade immune response and damage is the presence of numerous cytokines and cellular components in the niches of PCSCs, including non-PCSC cells, immune cells, inflammatory cells, vascular endothelial cells, and fibroblasts, as well as many growth factors and chemokines (Figure 4) [113].

Since PCSCs still belong to the category of cells in terms of the basic unit of life, the mechanism of PCSCs leading to radiotherapy resistance has varying degrees of intersection with other pathophysiological processes in this review. In the constructed radiation resistance model of PCa cells (LNCaP, DU145, and PC-3), markers of EMT are significantly upregulated, and the markers of CSCs (CD44, CD44v6, CD326, ALDH1, Nanog, and Snail) are also upregulated, suggesting the preliminary correlation between PCSCs and radiosensitivity. Additional investigation and analysis revealed a tight relationship between activation of the PI3K/Akt/mTOR pathways and the stem phenotype of PCa-resistant cells [114]. In ALDH+ PCSCs, the activated EMT process and reinforced DNA repair ability lead to PCa cells’ resistance to radiotherapy, which may be related to the promoted transcription of ALDH1A1 after the activation of the Wnt/β-catenin pathway. Moreover, the enhanced transcriptional expression of ALDH1 can eliminate ROS produced by oxidative stress in PCSCs and prevent genomic DNA damage [115]. SOX2, a well-known Yamanaka factor, is one of the core transcription factors for maintaining the pluripotent stemness of embryonic stem cells and CSCs and is also an important chemical promoting tumor development [116]. In PCa DU145 cells, the expression of SOX2 improves the anti-apoptotic ability by delaying caspase-3 cleavage, while knocking down SOX2 has the effect of radiation sensitization [117]. Another key marker of PCSCs is CD44v6. The inclusion of CD44v6 in PCa cells has been linked to a surge in cell proliferation, the formation of spheres, and resistance to various forms of chemotherapy, such as docetaxel, paclitaxel, doxorubicin, methotrexate, and even radiotherapy due to its involvement in the EMT process as well as activation of the PI3K/Ak/mTOR and Wnt/β-catenin pathways [118]. Structural maintenance of chromosomes 1A (SMC1A), a substrate molecule of ATM in response to DNA damage, is upregulated in PCa compared with normal tissues. After knockdown of SMC1A, the proliferation and sphere-forming ability of PCa DU145 and PC-3 cells decreased, and they became more sensitive to X-ray treatment, which was related to reversing the EMT phenotype and downregulating the stem cell markers (CD44, LEF-1, and POU5F-1) of PCSCs. SMC1A has been shown to enhance the efficiency with which HR and NHEJ can repair DNA damage. In addition, similar to ALDH1+ PCSCs, SMC1A can improve the antioxidant stress ability of cells through GSH and reduce the production of ROS [119]. The immune checkpoint B7-H3, a surface molecule on PCSCs, is considered a specific marker of PCSCs due to its significantly upregulated expression in the late stage of radiotherapy. Consequently, the development of chimeric antigen receptor T-cell (CART) therapies targeting this molecule can specifically target PCSCs, making PCa more responsive to radiation [120], showing promising applications for immunotherapy in the field of PCa. In essence, the radiotherapy tolerance of PCSCs is regulated by a wide range of molecules and processes such as surface molecular indicators, DNA damage repair, cell redox homeostasis regulation, signal pathway networks, and more. Identifying specific targets for PCSCs through scientific investigations could lead to the development of treatments that specifically target these cells.

## 8. Hypoxia in Tumor Core

When scientists first began researching the impact of radiation therapy on tumors, they initially concentrated on oxygen due to its significant influence on treatment outcomes when studying factors that affect the effects of radiation therapy. With advances in technology and a better understanding of oxygen, it was discovered that the oxygen partial pressure and oxygenation status in tumor tissues are important factors influencing the effectiveness of tumor treatments. The threshold of oxygen concentration that determines cell radiosensitivity is around 2%, beyond which X-rays can achieve more significant oxygen effects at lower concentrations. This has led to the development of the oxygen enhancement ratio (OER), a metric for comparing the relative radiation dose required to produce the same biological effect in an aerobic and oxygen-free environment. For radiation with low energy transfer (LET), such as X and γ-ray, the OER is about 2.5–3.5, while the OER of high-LET rays, such as proton and heavy-ion rays, is about 1.0 [121,122], which reflects that high-LET rays are less dependent on oxygen. X-rays are mainly used in current clinical routine radiotherapy. It is essential to have a clear understanding of the connection between tumor radiotherapy resistance and oxygenation status and its mechanism.

To put it another way, the “oxygen fixation hypothesis” suggests that X-rays can “fix” oxygen through indirect ionization to damage biological macromolecules [123]. Due to the anatomical position relationship between the tumor center and surrounding blood vessels, as well as abnormalities in blood vessels, the hypoxia zone, relative hypoxia zone, and normal oxygenation zone gradually form from the tumor center to the outside during the occurrence and progression of tumor cells (Figure 5). From the perspective of energy metabolism, the formation of hypoxia is due to an imbalance between oxygen acquisition and consumption. Currently, the hypoxic zone is considered one of the primary reasons for tumor radiochemotherapy resistance and a major characteristic of tumors. The hypoxic zone plays a critical role in different stages of tumor progression (including tumor cell proliferation, survival, angiogenesis, migration, cancer metabolic reprogramming, and stem cell characteristics) [124,125]. Targeting the hypoxic region in the center of the tumor is considered a therapeutic approach to alleviate treatment resistance and improve curative effects. PCa also exhibits hypoxia, which is also a hallmark of malignancies. PCa shares the characteristics of hypoxia with malignant tumors, and nitroimidazole compounds are among the earliest hypoxic sensitizers used clinically. Based on nitroimidazole compounds, a specific hypoxia probe named ^18^F-PEG3-ADIBOT-2NI-GUL targets the prostate-specific membrane antigen (PSMA) and can accurately display hypoxic regions within prostate cancer [126]. Carnitine palmitoyltransferase 1A (CPT1A) is one of the biomarkers for the β-oxidation of fatty acids. In conjunction with the nitroimidazole compound, pamomycin, it can be used for the fluorescent imaging of nude mice transplanted tumor models and provide new technologies and methods to display hypoxic regions in prostate cancer [127]. Several hypoxia probes are currently available, including ^18^F-[2-(2-nitro-1-H-imidazol-1-yl)-N-(2,2,3,3,3-pentafluoropropyl) acetamide (^18^F-EF5) [128], ^18^F-misonidazole (MISO) [129], and gadolinium tetraazacyclododecanetetraacetic acid monoamide conjugate of 2-nitroimidazole (GdDO3NI) [130]. Compared to the standard practice of inserting electrodes into tumor tissues for measuring oxygen partial pressure and hypoxia, these molecular probes have the advantages of being non-invasive, fast, and efficient. They also provide a valuable reference for outlining the biological lack of an oxygen target area in prostate cancer, making them a development trend in precision medicine. Additionally, researchers have modified the structure of nitroimidazole compounds and designed a novel type of ZIF-82-PVP nanoparticle material. By using radiotherapy X-ray controlled-release RNS, the apoptosis of hypoxic cells is increased while the protective autophagy of prostate cancer cells is significantly inhibited and nitrosation stress in PCa cells is boosted, thereby achieving targeted treatment for the hypoxic region of prostate cancer [131]. Under hypoxic conditions, liposome doxorubicin can reverse the vascular changes induced by hypoxia, greatly increasing PCa’s response to radiotherapy [117].

Tumor tissue hypoxia is a complex and spatiotemporal pathophysiological process. Analyzing the regulation mechanism of hypoxia in PCa is currently a research hotspot, as it can contribute to the discovery of corresponding targets and the development of sensitizers. Hypoxia-inducible factors (HIFs) are transcription factors that aid cancer cells in adapting to low-oxygen conditions by activating the transcription of several genes through binding with hypoxia response elements (HRE) in the promoter region. This process is crucial for maintaining the body’s oxygen homeostasis. The overexpression of HIF has been linked to immune escape, drug resistance, tumor neovascularization, metastasis, and tumor invasion and migration [132,133]. During the preliminary stages of research, small interfering RNA (siRNA) technology was used to knock down HIF-1 in PC-3 cells, revealing its effect on radiation sensitization [134]. Hormone-sensitive PCa LNCaP cells can also be treated with a HIF-1 inhibitor [135]. As our understanding of the underlying mechanisms of these processes grows, studies have demonstrated that HIF-1 expression in PCa cells can facilitate DNA repair and induce radiotherapy resistance by activating gene expression along the NHEJ-related pathway and facilitating the nuclear translocation of β-catenin [136]. HIF-1 can also co-transcribe with Nrf2 to control the expression of dimethylarginine dimethylaminohydrolase 1 (DDAH1) and thus enhance the development of PCa. Many small chemical inhibitors of HIFs, such as salicylic acid [137], manganese dioxide particles [138], statins [139], and metformin [140], have been studied for their potential to sensitize cells to radiation by alleviating the hypoxia of PCa tumors. HIFs are molecular targets for the radiosensitization of PCa, primarily connected by E3 ubiquitin ligase (such as VHL) and then degraded through the 26S ubiquitin-proteasome pathway. Proteolysis-targeting chimeras (PROTACs) [141] and molecular glue [142] technologies provide new chemical methods and schemes for the intracellular degradation of HIFs, worthy of exploration in the radiotherapy of PCa.

The oxygenation state of tumor tissue is different from that of normal tissue, providing a theoretical basis for the development of new clinical radiotherapy technology. New ultra-high-dose-rate radiation techniques, such as ultra-high-dose-rate (FLASH) radiotherapy, are being developed to achieve the goal of high-dose irradiation-resistant normal tissue while leaving the radiosensitivity of tumor tissue unaffected. FLASH treatment can provide a dosage of more than 8 Gy in a very short amount of time (often less than 1 s). The effect was first observed in 1959 [128]. Currently, there are two hypotheses regarding the biological mechanism of FLASH radiotherapy. One hypothesis is that under ultra-high-dose-rate radiotherapy, normal tissue becomes hypoxic due to the great consumption of oxygen, leading to resistance to radiation. The relatively hypoxic nature of tumor tissue makes it less affected by radiation with a high-dose rate, which promotes a “response error” between normal tissues and tumor tissues to IR. The other hypothesis is that the extremely short treatment time of FLASH can ensure the survival of circulating immune cells, thereby playing a role in systemic anti-tumor immunity. Preliminary explorations show that FLASH radiotherapy is closely related to hypoxia of tumor tissue; however, the specific biological effect of FLASH is yet to be determined, and its effectiveness needs to be clarified at the physical level in the advanced stages. Furthermore, mechanism exploration should be conducted on this basis to serve clinical practice.

## 9. Conclusions

Based on the research progress outlined above, the radiotherapy resistance of PCa has long been overlooked in clinical practice. In particular, the resistance to radiotherapy in highly malignant PCa requires more attention as investigations continue to advance. Radiotherapy resistance in PCa is a time-consuming and complex scientific issue that involves numerous biological and pathophysiological processes, such as defects and disorders of the DNA damage repair system, cell cycle disorder, imbalance of redox homeostasis, EMT, PCSCs, and hypoxia in the tumor core. Recently, immune factors have also been found to play a specific role in the radiotherapy resistance of PCa. To reverse or even revert the radiosensitivity of PCa and overcome radiotherapy resistance, the overall and molecular mechanisms of radiotherapy resistance in PCa need to be carefully analyzed. Corresponding targets to develop small-molecule inhibitors and immunotherapeutic drugs should be explored deeply, in the hopes of gaining fresh insights into overcoming clinical radiotherapy resistance (Appendix A).

## Figures and Tables

**Figure 1 biomedicines-11-01628-f001:**
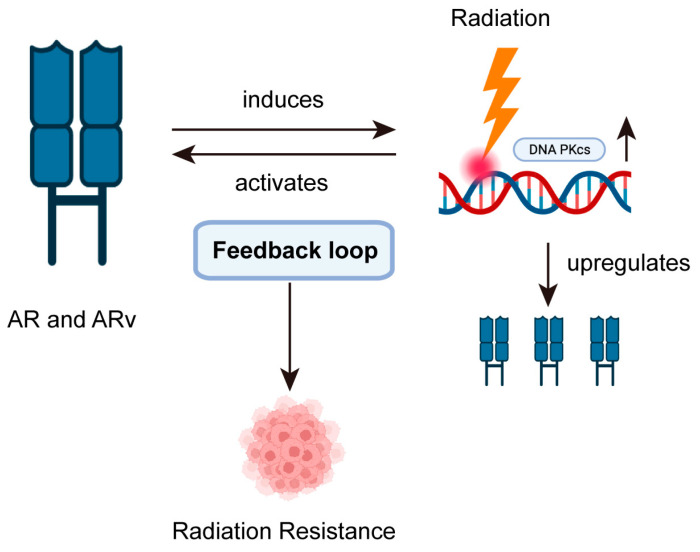
The complex regulatory link between AR and DNA PKcs.

**Figure 2 biomedicines-11-01628-f002:**
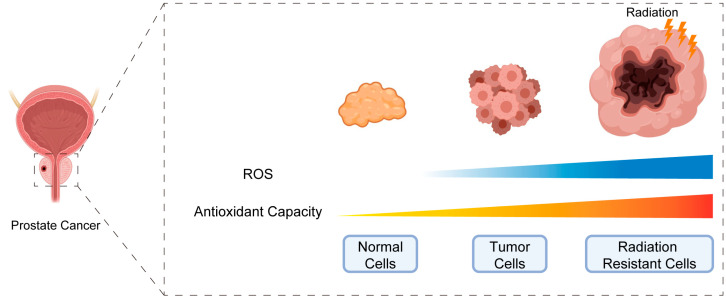
A schematic diagram showing the relationship between redox homeostasis and radiosensitivity.

**Figure 3 biomedicines-11-01628-f003:**
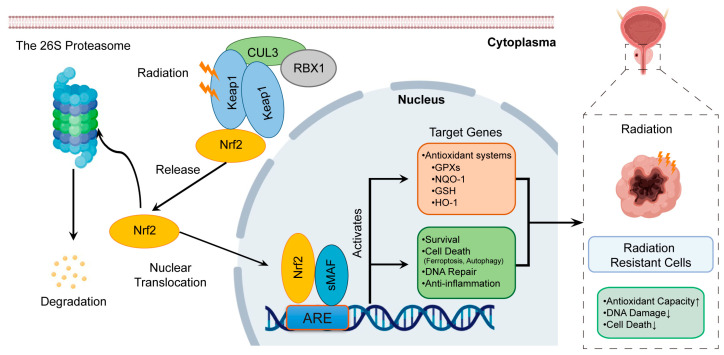
A schematic diagram of the “Nrf2/ARE” antioxidant stress signal pathway.

**Figure 4 biomedicines-11-01628-f004:**
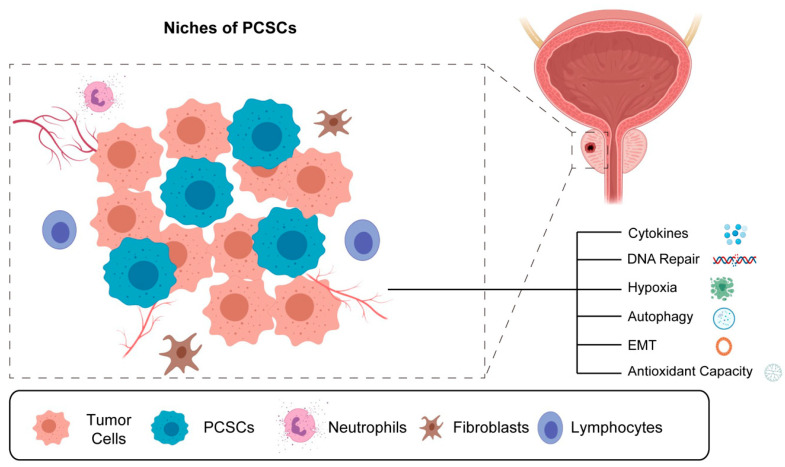
The niche and characteristic diagram of niches of PCSCs.

**Figure 5 biomedicines-11-01628-f005:**
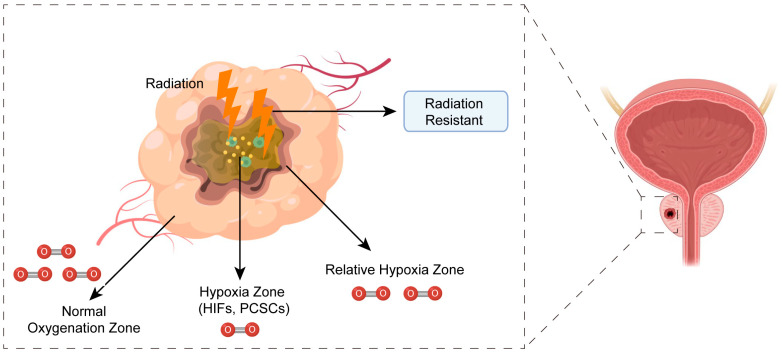
A schematic diagram that depicts the regions of hypoxic tumor tissue.

**Table 1 biomedicines-11-01628-t001:** The evolution of DNA damage repair pathways and radioresistance in PCa.

Types	Applicable Conditions	Relevant Research Progress
Base Excision Repair (BER)	A small amount of base damage	Soy isoflavones make PCa PC-3 cells more sensitive to radiation by inhibiting the expression of DNA repair enzyme APE1/Ref-1 in the nucleus [27]. XRCC1 R194W SNP information is a predisposing factor for PCa patients [28,29]. The presence of XRCC3 rs1799794 SNP information predicts the likelihood of gastrointestinal (GI) toxicity in PCa patients following radiotherapy [30].
Nucleotide Excision Repair (NER)	Larger pyrimidine dimers formed after excision of DNA damage	Numerous studies have investigated the SNPs in NER-related genes [31]. The PAT+/- polymorphism in the XPC gene is a predisposing factor for PCa susceptibility [32]. Cadmium exposure affects the expression of NER-related genes, including XPA [33]. The ERCC2 mutations (G > A, Asp (711) Asp) are predictive markers of toxic reactions to radiotherapy in PCa patients [34].
NHEJ	DNA double-strand break (DSB), independent of the cell cycle	The LIG4 (T > C, Asp (568) Asp) variant serves as a predictor of toxic reactions to radiotherapy in PCa patients [34]. The inhibitor of DNA PKcs and ATM, silymarin, accelerates the radiosensitivity of DU145 cells in PCa [35]. The expression level of Ku70 in PCa tissue can predict the treatment response of radiotherapy [36]. By targeting LITAF, miR-106 increases the expression of ATM, promoting radiotherapy resistance in PC-3 and DU145 cells of PCa [37]. The catalytic activity of Tip60 on ATM acetylation and phosphorylation in PCa cells makes it a candidate marker for radiotherapy resistance [38].
Homologous Recombination Repair (HRR)	DNA double-strand break (DSB), dependent on the existence of sister chromatids in the cell cycle	Approximately 30.7% of PCa patients harbor BRCA1/2 mutations [39]. In patients with mCRPC and BRCA1/2 or HRR mutations, the FDA has authorized the use of the PARP inhibitor Rucaparib [40]. Patients with mCRPC and the HRR mutation may benefit from Olaparib [41]. Silencing of RAD51 in PCa DU145 cells improves their radiosensitivity [42]. IL-6 promotes resistance to radiation in PCa C4-2 and 22Rv1 cells via DNA-damage-repair-related molecules (ATM, ATR, and BRCA1/2), and this effect can be counteracted with JAK and STAT3 inhibitors [43]. After ATM deletion, PCa cells (DU145, LNCaP, and 22Rv1) display increased sensitivity to radiation and ATR inhibition [44].
Cross-Link Repair	Cross-linking between DNA-DNA and DNA-protein due to IR	PCa harbors mutations in genes that encode the core complex of Fanconi anemia (FA), including FANCA ex1-12del and FANCA c.3384-1 G > A, but the relationship between these mutations and radiotherapy remains unexplored [45]. The S1088F mutant protein of FANCA enhances the susceptibility of cells to DNA damage induced by cis platinum [46].
Mismatch Repair (MMR)	Mismatch while removing replication and mismatch of small insertions	Although infrequent, the occurrence of MMR gene mutations in PCa serves as an unfavorable prognostic marker [47]. The level of MS1H6 expression is linked to Gleason Grade 5 [48]. PMS2 and MLH1 induce downregulation of BCL2A1- and c-Abl-mediated apoptosis in PCa DU145 cells, suggesting their potential as targets for radiosensitization [49,50]. Mutations in MLH1 and PMS1 impact the sensitivity of Olaparib [51].

**Table 2 biomedicines-11-01628-t002:** A summary of the improvements in radiotherapy that have been accomplished through targeting cell cycle regulatory molecules.

Target	Phase	Molecular Mechanism
miR-106b [53]	G2/M arrest	miR-106b can inhibit the proliferation of the PCa LNCaP cells by activating P21-mediated cell cycle arrest.
The receptor of Exendin-4, namely, GLP-1R [54]	G2/M arrest	Exendin-4 promotes AMPK phosphorylation and activates downstream signaling pathways, while also inhibiting the expression of mTOR, cyclinB, and p34.
miR-449a [55]	Cdc25A, Cdc2/CyclinB	Induction of G2/M arrest.
CD105 [56]	G2/M arrest	CD105 has been shown to promote radiotherapy resistance in PCa by depleting intracellular ATP and upregulating SIRT1, which activates the BMP and TGF-β/Smad pathways. However, targeting CD105 with the TRC105 antibody has been demonstrated to enhance radiosensitivity.
Resveratrol [57]	G1/S arrest	Resveratrol has been shown to inhibit the phosphorylation of PI3K/Akt, which is an important cell survival signaling pathway in PCa 22Rv1 and PC-3 cells following radiation treatment. Additionally, resveratrol induces the phosphorylation of AMPK and promotes cell cycle arrest in a P21-dependent manner.
GnRHR [58]	G2/M arrest	Redistribution of GnRHR expression using IN3 repositioned it on the membrane surface of PC-3 cells, promoting their radiosensitivity in the recoverable phase, while IN3 had a pro-apoptotic effect.
RPS6KB1 [59]	cyclinD1, Cdc25C, G2/M arrest	The expression of ChK1, p-Cdc25C, and cyclinD1 in PCa PC-3 cells can be downregulated after RPS6KB1 is inhibited by Nexrutine. When RPS6KB1 is inhibited, and the process of NHEJ is also inhibited.
miR-16-5p [60]	Cyclin D1/Cyclin E1/pRb/E2F1	MiR-16-5p has been shown to downregulate cyclinD1 and E1 expression in PCa LNCaP cells by directly binding with their 3′UTR region, leading to cell cycle arrest at the G0/G1 phase.

## Data Availability

If necessary, the corresponding data for this review can be obtained by contacting the corresponding author (doctorgaoxs@126.com).

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
