# Peer review of "Uncovering the Secrets of Prostate Cancer’s Radiotherapy Resistance: Advances in Mechanism Research"

_biomedicines, 2023, doi:10.3390/biomedicines11061628_

Round 1

Reviewer 1 Report

This paper represents an extensive review of the  mechanisms underlying radiation therapy resistance in PCa.

On  Page 8/20 the authors introduce PSA. I recommend to introduce at least one sentence in the introduction (lines26-27)to explain that PSA is the reference marker for PCa diagnosis and monitoring, and the current position of the current guideline on treating only advanced PCa. This may be relevant for the aims of the paper. All these concepts may be found in this paper:

Ferraro S, Biganzoli D, Rossi RS,  Palmisano F, Bussetti M, Verzotti E, Gregori A, Bianchi F, Maggioni M,  Ceriotti F, Cereda C, Zuccotti G, Kavsak  P, Plebani M, Marano G, Biganzoli EM. Individual Risk Prediction of Advanced Prostate Cancer based on the combination between total Prostate-Specific Antigen (PSA) and free to total PSA ratioClin Chem Lab Med. 2023 Jan 27.

Minor corrections:

FLASH should be spell out in the Abstract and in the text

Later stages should be replaced by advanced stages

minor corrections

Author Response

Point-by-Point Response to Reviewer 1

Manuscript ID: biomedicines-2376072

Title: Uncovering the Secrets of Prostate Cancer's Radiotherapy Resistance: Advances in Mechanism Research

NOTE: The texts in Bold are the reviewers' comments. The revised sentences are marked with red font in the manuscript. As images cannot be uploaded in the text box, the image section of responding to the reviewer's comments can be downloaded as an attachment.

Thank you very much for providing us with constructive comments on our manuscript. We sincerely appreciate your efforts in helping us to improve our work. We have thoroughly revised the manuscript according to your valuable feedback and have addressed all of your concerns. We hope that these adjustments have satisfactorily addressed any issues raised during the review process.

Revision 1: We added background knowledge of PSA in the introduction section to help readers better understand prostate cancer.

Revision 2: We added a graphic discussion on the correlation between variant histology and radiotherapy in prostate cancer at the EMT section, further enriching the content of the clinical part of the manuscript and making it more clinically relevant.

Revision 3: We added supplementary Figure 1 to show the time span and important biological events included in the reference list of the manuscript to clearly demonstrate the important biological discoveries and milestones related to radiotherapy resistance in the field of prostate cancer.

Revision 4: We added supplementary Figures 2 and 3 to better illustrate the specific role of relevant molecules, bio-compounds, and small molecules in radiotherapy resistance in prostate cancer.

Revision 5: We corrected some language errors.

Question 1: On Page 8/20 the authors introduce PSA. I recommend to introduce at least one sentence in the introduction (lines 26-27) to explain that PSA is the reference marker for PCa diagnosis and monitoring, and the current position of the current guideline on treating only advanced PCa. This may be relevant for the aims of the paper. All these concepts may be found in this paper:

Ferraro S, Biganzoli D, Rossi RS, Palmisano F, Bussetti M, Verzotti E, Gregori A, Bianchi F, Maggioni M, CeriottiF, Cereda C, Zuccotti G, Kavsak P, Plebani M, Marano G, Biganzoli EM. Individual Risk Prediction of Advanced Prostate Cancer based on the combination between total Prostate-Specific Antigen (PSA) and free to total PSA ratioClin Chem Lab Med. 2023 Jan 27.

Response:

The comments are very helpful. Thanks for your comments and we have added corresponding sentence and reference according to the Reviewer 's suggestions as follows: Prostate-specific antigen (PSA) is a prevalent biomarker employed for the diagnosis and active surveillance of PCa. According to the NCCN guidelines, PSA plays a crucial role in determining PCa risk stratification. Specifically, a PSA level of less than 10 ng/mL indicates low-risk PCa, 10 to 20 ng/mL suggests intermediate-risk PCa, and levels exceeding 20 ng/mL are indicative of high-risk PCa. Consequently, PSA screening is an important component for the diagnosis of PCa (lines 26 to 31 in revised manuscript).

Reference:

Ferraro, S.; Biganzoli, D.; Rossi, R.S.; Palmisano, F.; Bussetti, M.; Verzotti, E.; Gregori, A.; Bianchi, F.; Maggioni, M.; Ceriotti, F.; et al. Individual risk prediction of high grade prostate cancer based on the combination between total prostate-specific antigen (PSA) and free to total PSA ratio. Clin Chem Lab Med 2023, doi:10.1515/cclm-2023-0008.

Question 2: FLASH should be spell out in the Abstract and in the text.

Response:

The comments are very helpful. Thanks for your comments and we are very sorry for our negligence of this words mistake. We have re-written this sentence according to the Reviewer 's suggestions as follows:

  1. This can also provide a theoretical basis for the clinical application of novel ultra-high dose rate (FLASH) radiotherapy in the era of PCa (lines 18 to 19 in revised manuscript).
  2. New ultra-high dose rate radiation techniques, such as ultra-high dose rate (FLASH) radiotherapy (lines 505 to 506 in revised manuscript).

Question 3: Later stages should be replaced by advanced stages.

Response:

The comments are very helpful. Thanks for your comments and we are very sorry for our negligence of this words mistake. We have re-written this sentence according to the Reviewer 's suggestions as follows: the specific biological effect of FLASH has yet to be determined, and its effectiveness needs to be clarified at the physical level in the advanced stages (line 520 in revised manuscript).

We greatly appreciate your efforts in reviewing our work and understand the importance of constructive criticism in enhancing the quality of our research. Your feedback has been instrumental in guiding us towards the best possible outcome.

Thank you once again for your time and expertise. It has been a pleasure working with you.

Reviewer 2 Report

As consultant urological surgeon I have read with interest this manuscript by Feng Lyu and colleagues.

The paper provides some important insights about PCa's radiotherapy Resistance particularly focusing on the molecular and related genetic basis. The authors mentioned the importance of epithelial-mesenchymal transition (EMT) in this clinical scenario. EMT is further related to molecular divergent subtypes and aberrant histologies. For a more clinical perspective I would add an additive paragraph about the variant ("subtypes" as per the novel WHO 2022 edition - doi: 10.1016/j.eururo.2022.07.002) histologies. Variant histologies have been reported as drivers of biological heterogenicity and related aggressiveness. The authors should refer to some important paper in the field of uro-oncology (NMIBC - doi: 10.1007/s00428-021-03264-6; MIBC - doi: 10.1111/bju.15984) and further prostate cancer (doi: 10.1097/PAI.0000000000001067; doi: 10.1038/s41585-021-00447-3; doi: 10.1016/j.clgc.2023.03.010) and PCa patterns such as cribriform as well (doi: 10.1038/s41391-022-00600-y).

The authors should further add a "Methods" section to briefly describe the timeframe and the settings of the search strategy.

Author Response

Point-by-Point Response to Reviewer 2

Manuscript ID: biomedicines-2376072

Title: Uncovering the Secrets of Prostate Cancer's Radiotherapy Resistance: Advances in Mechanism Research

NOTE: The texts in Bold are the reviewers' comments. The revised sentences are marked with red font in the manuscript. As images cannot be uploaded in the text box, the image section of responding to the reviewer's comments can be downloaded as an attachment.

Thank you very much for providing us with constructive comments on our manuscript. We sincerely appreciate your efforts in helping us to improve our work. We have thoroughly revised the manuscript according to your valuable feedback and have addressed all of your concerns. We hope that these adjustments have satisfactorily addressed any issues raised during the review process.

Revision 1: We added background knowledge of PSA in the introduction section to help readers better understand prostate cancer.

Revision 2: We added a graphic discussion on the correlation between variant histology and radiotherapy in prostate cancer at the EMT section, further enriching the content of the clinical part of the manuscript and making it more clinically relevant.

Revision 3: We added supplementary Figure 1 to show the time span and important biological events included in the reference list of the manuscript to clearly demonstrate the important biological discoveries and milestones related to radiotherapy resistance in the field of prostate cancer.

Revision 4: We added supplementary Figures 2 and 3 to better illustrate the specific role of relevant molecules, bio-compounds, and small molecules in radiotherapy resistance in prostate cancer.

Revision 5: We corrected some language errors.

Question 1: The paper provides some important insights about PCa's radiotherapy Resistance particularly focusing on the molecular and related genetic basis. The authors mentioned the importance of epithelial-mesenchymal transition (EMT) in this clinical scenario. EMT is further related to molecular divergent subtypes and aberrant histologies. For a more clinical perspective I would add an additive paragraph about the variant ("subtypes" as per the novel WHO 2022 edition - doi: 10.1016/j.eururo.2022.07.002) histologies. Variant histologies have been reported as drivers of biological heterogenicity and related aggressiveness. The authors should refer to some important paper in the field of uro-oncology (NMIBC - doi: 10.1007/s00428-021-03264-6; MIBC - doi: 10.1111/bju.15984) and further prostate cancer (doi: 10.1097/PAI.0000000000001067; doi: 10.1038/s41585-021-00447-3; doi: 10.1016/j.clgc.2023.03.010) and PCa patterns such as cribriform as well (doi: 10.1038/s41391-022-00600-y).

Response:

Thank you for your comments. We appreciate your feedback regarding the lack of discussion on the correlation between EMT chapters and variant histologies in our manuscript. Your input is invaluable to the improvement of the quality of our work. As a result, we have carefully reviewed the literature you provided us, as well as literature related to variant histologies and prostate cancer radiotherapy, and have added the following content to our manuscript:

EMT is also associated with molecularly divergent subtypes and aberrant histologies. Variant histologies (VHs) have been recognized as drivers of biological heterogeneity and increased aggressiveness in the current clinical practice. In non-muscle-invasive bladder cancer (NIMBC), variant histologies (nested, glandular, micropapillary, squamous, inverted, basaloid, microcystic, villous-like, lymphoepithelioma-like carcinoma) have been identified as risk factors for patient disease-free survival (DFS) [98]. Plasmacytoid, small-cell, and sarcomatoid VHs are linked to worse disease-specific survival (DSS) in muscle-invasive bladder cancer (MIBC), while lymphoepithelioma-like VH is associated with an improved DSS [99]. Accurate pathological diagnosis of VHs can enable tailored counseling to identify patients who require more intensive management [100]. In addition, ductal adenocarcinoma (DAC) is the most common variant histological subtype of PCa and is characterized by an aggressive clinical course. Recent studies suggest that DAC requires external beam radiation therapy and particle-enhanced therapy, indicating DAC's resistance to radiation therapy [101,102]. Intraductal carcinoma of the prostate (IDC-P) is positively correlated with higher GS and is associated with early relapse and metastasis after radiation therapy, suggesting IDC-P's insensitivity to radiation therapy [103,104]. Sarcomatoid carcinoma is also rare and carries a poor prognosis, with limited clinical interventions and approximately 38% of patients experiencing distant metastasis [105]. It most frequently emergence after radiation for a high-grade acinar carcinoma [106]. Some sarcomatoid carcinomas lack classical epithelial features [107], which could be one of the reasons why these patients are resistant to chemotherapy and radiation therapy, leading to a poor prognosis. Besides, pleomorphic giant cell adenocarcinoma is a rare and aggressive subtype that often develops following prior treatment with androgen deprivation or radiation [108]. Therefore, these variant histologies are not only strongly associated with risk stratification and survival outcome in patients but also pose as a significant challenge in understanding the relationship and mechanism between ionizing radiation and specific pathological types in PCa (lines 329 to 355 in revised manuscript).

Thank you for your valuable feedback. Your suggestion has been particularly helpful in linking the unique clinical pathological subtype with prostate cancer radiotherapy. We appreciate your insightful input again.

Question 2: The authors should further add a "Methods" section to briefly describe the timeframe and the settings of the search strategy.

Response:

Thank you for your valuable feedback. Your suggestions have greatly helped to improve the quality of our manuscript. We acknowledge that the review covers a wide time span, and therefore we have included a timeline in Supplementary Figure 1 to highlight the milestones of relevant advancements in biology and radiotherapy technologies related to our review. This will help readers better understand the past, present, and future of prostate cancer radiotherapy resistance.

Supplementary Figure 1:

Supplementary Figure 1. Timeline of important radiotherapy techniques and biological processes studied in radiotherapy resistance of prostate cancer.

Regarding the issue of search strategy, as our review focused on the topic of prostate cancer radiotherapy resistance and was not a systematic review, there was no fixed search strategy. However, we did mainly focus on recent research progress, such as immunotherapy and FLASH radiotherapy. Therefore, our search was flexible and not fixed.

We greatly appreciate your efforts in reviewing our work and understand the importance of constructive criticism in enhancing the quality of our research. Your feedback has been instrumental in guiding us towards the best possible outcome.

Thank you once again for your time and expertise. It has been a pleasure working with you.

Reviewer 3 Report

This manuscript is well written and well organized. Just one suggestion for the authors. It could be better to illustrate some targeting therapeutic agents or bio-activate compounds that summarized in table 1 and 2 in the following figures. That will let audiences easy understand that each key regulator being targeted by which drugs or therapeutic strategies.

The scientific English written is good.

Author Response

Point-by-Point Response to Reviewer 3

Manuscript ID: biomedicines-2376072

Title: Uncovering the Secrets of Prostate Cancer's Radiotherapy Resistance: Advances in Mechanism Research

NOTE: The texts in Bold are the reviewers' comments. The revised sentences are marked with red font in the manuscript. As images cannot be uploaded in the text box, the image section of responding to the reviewer's comments can be downloaded as an attachment.

Thank you very much for providing us with constructive comments on our manuscript. We sincerely appreciate your efforts in helping us to improve our work. We have thoroughly revised the manuscript according to your valuable feedback and have addressed all of your concerns. We hope that these adjustments have satisfactorily addressed any issues raised during the review process.

Revision 1: We added background knowledge of PSA in the introduction section to help readers better understand prostate cancer.

Revision 2: We added a graphic discussion on the correlation between variant histology and radiotherapy in prostate cancer at the EMT section, further enriching the content of the clinical part of the manuscript and making it more clinically relevant.

Revision 3: We added supplementary Figure 1 to show the time span and important biological events included in the reference list of the manuscript to clearly demonstrate the important biological discoveries and milestones related to radiotherapy resistance in the field of prostate cancer.

Revision 4: We added supplementary Figures 2 and 3 to better illustrate the specific role of relevant molecules, bio-compounds, and small molecules in radiotherapy resistance in prostate cancer.

Revision 5: We corrected some language errors.

Question 1: This manuscript is well written and well organized. Just one suggestion for the authors. It could be better to illustrate some targeting therapeutic agents or bio-activate compounds that summarized in table 1 and 2 in the following figures. That will let audiences easy understand that each key regulator being targeted by which drugs or therapeutic strategies.

Response:

Thank you very much for your valuable comments. Your insights have been incredibly helpful to our team. In response to your feedback, we have made significant improvements in our manuscript. Previously, we had only used verbal descriptions to explain the targeted therapeutic agents or bio-activated compounds summarized in tables 1 and 2. Unfortunately, our other figures lacked detail and failed to clearly demonstrate the specific link between these natural compounds and small molecule inhibitors in DNA damage repair and cell cycle processes.

With great appreciation for your thoughtful suggestions, we have completely revised those figures and incorporated them into the supplemental materials. Thank you again for your guidance on this matter.

Supplementary Figure 2:

Supplementary Figure 2. Targeted therapeutic agents and bio-activated compounds: unveiling their specific role links in the DNA damage and repair pathways.

Supplementary Figure 3:

Supplementary Figure 3. Targeted therapeutic agents and bio-activated compounds: unveiling their specific role links in the cell cycle.

We greatly appreciate your efforts in reviewing our work and understand the importance of constructive criticism in enhancing the quality of our research. Your feedback has been instrumental in guiding us towards the best possible outcome.

Thank you once again for your time and expertise. It has been a pleasure working with you.

Round 2

Reviewer 1 Report

No further comments